# Analysis of Corneal Deformation in Paediatric Patients Affected by Maturity Onset Diabetes of the Young Type 2

**DOI:** 10.3390/diagnostics13081500

**Published:** 2023-04-21

**Authors:** Michele Lanza, Enza Mozzillo, Rosa Boccia, Ludovica Fedi, Francesca Di Candia, Nadia Tinto, Paolo Melillo, Francesca Simonelli, Adriana Franzese

**Affiliations:** 1Multidisciplinary Department of Medical Surgical and Dental Specialties, University of Campania Luigi Vanvitelli, 80138 Naples, Italy; 2Department of Translational Medical Sciences, Regional Center of Pediatric Diabetes, Federico II University of Naples, 80131 Naples, Italy; 3Department of Molecular Medicine and Medical Biotechnology, CEINGE Advanced Biotechnologies, Federico II University of Naples, 80131 Naples, Italy

**Keywords:** MODY2, diabetes, cornea, biomechanics, Corvis ST

## Abstract

Background: To evaluate corneal deformation in Maturity Onset Diabetes of the Young type 2 (MODY2), paediatric subjects were analysed using a Scheimpflug-based device. The purpose of this analysis was to find new biomarkers for MODY2 disease and to gain a better understanding of the pathogenesis of the disease. Methods: A total of 15 patients with genetic and metabolic diagnoses of MODY2 (mean age 12.8 ± 5.66 years) and 15 age-matched healthy subjects were included. The biochemical and anthropometric data of MODY2 patients were collected from clinical records, and a complete ophthalmic check with a Pentacam HR EM-3000 Specular Microscope and Corvis ST devices was performed in both groups. Results: Highest concavity (HC) deflection length, Applanation 1 (A1) deflection amplitude, and A1 deflection area showed significantly lower values in MODY2 patients compared to healthy subjects. A significant positive correlation was observed between Body Mass Index (BMI) and HC deflection area and between waist circumference (WC) and the following parameters: maximum deformation amplitude, HC deformation amplitude, and HC deflection area. The glycosylated hemoglobin level (HbA1c) showed a significant positive correlation with Applanation 2 time and HC time. Conclusions: The obtained results show, for the first time, differences regarding corneal distortion features in the MODY2 population compared with healthy eyes.

## 1. Introduction

Maturity Onset Diabetes of the Young type 2 (MODY2) is the most common form of monogenic diabetes and is caused by heterozygous loss-of-function glucokinase (GCK) gene mutations that encode the glucokinase enzyme. The loss of GCK function has negative effects on the kinetic parameters of the enzyme, leading to a glucose-sensing defect and, therefore, an increase in the blood glucose threshold that triggers insulin secretion [1,2,3,4]. Clinically, MODY2 is characterised by a mild increase in fasting glucose and the glycosylated hemoglobin (HbA1c) level without the need for pharmacological intervention, except during pregnancy [5,6]. Due to the mild phenotype, it is evident that MODY2 patients require less frequent clinical surveillance than patients with other forms of diabetes, though this is also because neither micro- nor macrovascular complications are involved [1,2,3]. However, our current knowledge of MODY2 is quite recent, having only been established over the last two decades, and as a result, little or nothing is known about MODY2’s ocular complications, which require further exploration. Moreover, there is a lack of long-term data on the organs that, in the most common forms of diabetes, suffer from complications. In fact, diagnoses are generally made at a young age (8–14 years), but no significant data on follow-ups have been published. Child diagnoses can also lead to disease identification in a parent or even in a grandparent [5,7]. In these cases, the search for complications could provide even more interesting data. Complications in the most common forms of diabetes include morphological, glucometabolic, and functional changes in the cornea. Hyperglycemia, which is typically associated with the disease, increases protein glycosylation and induces corneal collagen crosslinking, resulting in biomechanical alterations [8,9].

Corneal biomechanics comprises a group of new parameters involved in corneal deformation and has been studied in relatively recent times [10]. The first device for measuring corneal distortion was the ocular response analyser (ORA), (Reichert Inc., Depew, NY, USA); subsequently, another instrument for measuring different parameters was released. This device measures the corneal response to an external air puff stimulus recorded by a Scheimpflug camera [11,12]. Different studies focusing on corneal biomechanics involving both healthy eyes and eyes affected by ocular and systemic diseases have been published. They aimed to better understand the mechanisms underlying corneal alterations and to find new biomarkers for early identification and better management of corneal deformation [13,14,15]. Some of these studies, which were conducted in type 2 adult diabetic patients, provided interesting results suggesting that corneal biomechanics could be useful in the overall care of these patients [16,17,18,19]. Conversely, studies that focused on children with type 1 diabetes and only used an ORA showed no statistical difference in corneal deformation parameters compared with healthy subjects [20,21]. Having reliable parameters could be useful for clinicians to better identify MODY2 patients, whose diagnosis is difficult and is often confused with other forms of diabetes. Moreover, unveiling some hidden characteristics of the cornea could help us to better understand the pathogenesis of the disease and improve its management. No data have been published on corneal distortion evaluation in paediatric patients with monogenic diabetes. Among the few studies evaluating corneal deformations in paediatric diabetic patients, this is the first one to assess corneal deformations in MODY2 paediatric subjects via a Scheimpflug-based device.

## 2. Materials and Methods

In this prospective, comparative study, Caucasian children and adolescents were included. Group 1 comprised 15 MODY2 subjects, and Group 2 comprised 15 age-matched healthy subjects. Healthy subjects were selected from the paediatric outpatients of the Eye Unit of the University of Campania Luigi Vanvitelli. MODY2 patients were consecutively enrolled in the Regional Center of Paediatric Diabetes of Federico II University Hospital and tested for mutations in the GCK gene at the molecular diagnostics laboratory for monogenic diabetes of the CEINGE Advanced Biotechnology.

The study protocol was approved by the local Ethics Committee of the University of Campania Luigi Vanvitelli and was conducted in accordance with the Helsinki Declaration and Good Clinical Practice guidelines. Informed written consent was obtained from all parents and subjects prior to study inclusion.

Medical history, biochemical parameters, and clinical data were obtained from clinical records, and they are shown in Table 1.

Inclusion criteria for both groups were availability of glucose control and anthropometry data, while inclusion criteria for diabetic subjects were genetic and metabolic data regarding MODY2 diagnosis [3]. Exclusion criteria were a best-corrected visual acuity worse than 20/20 for both groups and presence of ocular pathology and chronic and/or systemic disease except MODY2 for Group 1.

All subjects included in the study underwent a complete eye visit with visual acuity and refraction test, biometric and corneal evaluation using Pentacam HR (Oculus, Wetzlar, Germany) and EM-3000 Specular Microscope (Tomey Corporation, Nagoya, Japan), Corvis ST scans, and fundus examination using indirect ophthalmoscopy.

Pentacam HR is a corneal tomograph that utilises a rotating Scheimpflug camera and a monochromatic slit light source (blue LED at 475 nm) that rotate together around the optical axes of the eye to calculate a three-dimensional model of the anterior segment that includes data from anterior and posterior corneal topography, pachymetry, and measurements of anterior chamber depth, lens opacity, and lens thickness. Within 2 s, the system rotates 180° and acquires 25 or 50 images (depending on the user’s settings) that contain 500 measurement points on the front and back corneal surfaces to draw a true elevation map [22]. For this study, the 25-images-per-scan option was chosen. Central corneal thickness (CCT) and keratometry provided by this device were included in this study to evaluate the morphological changes.

Corvis ST uses 4330-images-per-second Scheimpflug camera to record corneal behavior during an air puff indentation. It provides many different parameters related to the main moments: the first applanation occurs when cornea flattens because of the air puff, and the second applanation occurs when the cornea flattens again after reaching the highest concavity shape in order to return to its original shape. The first applanation (A1) parameters are first applanation time, first applanation length, and first applanation velocity. The second ones (A2) are second applanation time, second applanation length, and second applanation velocity. Features obtained when the cornea reaches the highest concavity (HC) position are highest concavity time, highest concavity peak distance, highest concavity peak radius, and highest concavity deformation amplitude [23]. All parameters provided by Corvis ST have been included in the analysis.

For each study participant, the following data were recorded: age, sex, height, and weight; for diabetic patients, time elapsed since diagnosis of MODY2, fasting blood glucose, waist circumference, and the last HbA1c were also recorded; fasting blood glucose was also measured in controls in order to exclude any form of diabetes. Weight and height were collected by a single trained operator. HbA1c was assessed by High Performance Liquid Chromatography.

Ocular parameters included in the analysis were visual acuity, refraction, endothelial cell density (ECD), keratometry, anterior chamber depth (ACD), intraocular pressure (IOP) measured with Corvis ST, CCT, corneal volume (CV), and, in addition, all parameters provided by Corvis ST related to corneal distortion. Both eyes of same subjects were included in the evaluations.

### Statistical Analysis

Linear regression, estimated by a generalised estimating equation (GEE), was fitted on the data to investigate differences between the two groups. Furthermore, regression models were fitted to explore the relationship between the selected variables in Group 1; GEE was applied since this method could accommodate the inter-eye correlation. A *p*-value of <0.05 was considered significant.

## 3. Results

As shown in Table 1, age, sex, and refractions were similar in both groups. In Table 2, it is possible to observe auxological data in both groups and other parameters related to the diabetic patients.

A comparison of the features evaluated by a Specular Microscope, Corvis ST, and Pentacam is shown in Table 3 and Table 4. As for the morphological data, CCT and CV were found to be thicker in healthy subjects, regardless of the device that was used (Specular Microscope, Corvis ST, and Pentacam). Higher values of ECD were observed in MODY2 patients, whereas no significant differences were detected when evaluating IOP and corneal curvature (Table 3).

Table 4 shows a comparison of the biomechanical data obtained by Corvis ST in MODY2 patients and healthy subjects. It is possible to observe that HC deflection length, A1 deflection amplitude, and A1 deflection area showed significantly lower values in MODY2 patients compared to healthy subjects, suggesting reduced corneal deformability of the eyes in the MODY2 group. Interestingly, the morphological features of MODY2 patients showed lower CCT and CV values, which are usually associated with higher corneal deformability (Table 3).

Further analysis involved the evaluation of the correlations among biomechanical characteristics of MODY2 corneas and systemic parameters (Table 5). This analysis showed that, among the parameters with a significant difference between healthy and MODY2 eyes, only HC deflection length provided a significant positive correlation with waist circumference, whereas A1 deflection amplitude and A1 deflection area did not. Moreover, significant positive correlations were observed between waist circumference and the following parameters: maximum deformation amplitude, HC deformation amplitude, and HC deflection area. A significant positive correlation was observed between Body Mass Index (BMI) and HC deflection area. HbA1c showed a significant positive correlation with A2 time and HC time.

## 4. Discussion

In adult patients affected by diabetes mellitus, corneal biomechanical proprieties have been proven to show significant differences from healthy people.

The systematic review and meta-analysis provided by Wang et al. produced an evaluation of 15 studies measuring corneal biomechanical properties and intraocular pressure with an ORA in a total of 1506 eyes in the diabetic group and 2190 eyes in the control group. After an extensive analysis, the authors reported that higher values of both corneal hysteresis and corneal resistance factors are related to diabetes mellitus, but both IOP parameters provided by an ORA are higher in diabetic patients [13].

Del Buey et al. published a very interesting review about corneal structure changes in diabetic patients. In regard to corneal biomechanical properties, they reported that the majority of studies detected an increase in corneal hysteresis values, provided by an ORA, in diabetic patients compared to matched population groups without this disease. It is important to highlight that there were other studies reporting lower values of corneal hysteresis or no significant differences. Thus, the discussion about the influence of diabetes on corneal distortion properties is still open, with several aspects left to unveil [16].

In their observational, cross-sectional, observer-masked study, Perez-Rico et al. evaluated one eye of 94 consecutive diabetic patients and 41 healthy subjects. Moreover, they divided the diabetic participants into controlled and uncontrolled groups. Corneal biomechanical properties were assessed using both an ORA and Corvis ST, and then the values obtained were correlated with corneal thickness. According to their data, poor glucose control is associated with lower corneal hysteresis and lower Corvis ST parameters related to corneal deformation [17].

In 2019, Ramm et al. assessed corneal deformation parameters, provided by both an ORA and Corvis ST, in 35 diabetic patients and 35 healthy subjects. They found higher values of both corneal hysteresis and corneal resistance factors provided by the ORA in diabetic patients. While evaluating corneal biomechanical properties using the Corvis ST, the authors noticed that only A1 and A2 deflection amplitudes were increased (*p* < 0.001), and HC and A2 time were extended in diabetic patients compared to healthy subjects [18].

Ramm et al. evaluated corneal biomechanical properties using both an ORA and Corvis ST in 81 diabetic patients and 75 healthy participants. They found several significant differences between the parameters provided by the two devices. In particular, corneal hysteresis was higher in diabetic corneas, and so were HC, A2 time, and A1 and A2 deflection amplitudes. Moreover, a correlation analysis between the severity of diabetic retinopathy and the level of corneal biomechanical modifications was conducted. The results suggested developing a compensative diabetic index to increase the reliability of corneal deformation analysis [19].

These findings are mainly related to metabolic disorders leading to a change in corneal structure and, consequently, corneal deformation. However, little research has been conducted on the paediatric population. Moreover, in previous studies on children, only an ORA device was used to measure the corneal biomechanical properties without detecting significant differences between diabetic and non-diabetic patients [20,21].

Even if the literature does not report univocal findings, numerous authors evaluated the differences in CCT, measured with several devices, when comparing diabetic and non-diabetic populations and noted a greater central corneal thickness in the first group [24,25,26,27,28].

According to data published in different studies, CCT values in the paediatric population are strongly influenced by HbA1c levels with a positive correlation, even if they are measured with different devices [29,30,31,32].

On the contrary, in this study, lower CCT values were observed in MODY2 patients, even if the difference detected (15.07 µm) was not clinically relevant. Thus, this is an uncommon result and could be related to the accuracy limit of the device used to perform CCT measurements, even if Pentacam HR has an accuracy of 3.77 µm with a minimum detectable difference of 1 µm [33]. Another possible explanation could be the fact that most of the population with MODY2 has HbA1c levels slightly above the normal limit, which might not determine an increase in corneal thickness.

According to the data observed in this study, ECD was higher in MODY2 patients compared to healthy subjects (Table 3). Previous studies evaluating the endothelial function in adult diabetic patients (both insulin-dependent and non-insulin-dependent) showed controversial results: some of them reported lower cell counts or increased pleomorphism, whereas others did not detect anomalies [27,34,35,36]. The different results observed in the current study could be related to the different kinds of diabetes evaluated in previous papers, considering that no paediatric subjects were analysed and that patients included in this study did not show significant glycemic alterations.

Previous research reported an increase in intraocular pressure (IOP) in adult diabetic patients, but no details are provided about this comparison in paediatric ones [37,38].

Scheler A et al. measured IOP using an ORA, Goldmann Applanation Tonometry, and Dynamic Contour Tonometry in 35 eyes of healthy subjects and 31 eyes of diabetic adult patients. They found that IOP values provided both by ORA and Goldmann Applanation Tonometry were significantly higher in patients affected by diabetes mellitus compared to healthy eyes, and those provided by Dynamic Contour Tonometry showed a tendency to increase in diabetic population, even if the difference was not significant [37].

Ramm et al. reported higher values of IOP measured with Goldmann Applanation Tonometry, Corvis ST, and an ORA in adult patients affected by diabetes mellitus compared to healthy eyes [38].

In this study, the IOP values, which were measured with Corvis ST without compensating for corneal biomechanics, were significantly higher in the control group. However, a bIOP level analysis accounting for corneal distortion showed, for the first time, a significantly higher value in the affected paediatric population. The bIOP data are an expression of IOP corrected by biomechanical properties. In fact, this study showed that some biomechanical characteristics, which were analysed with Corvis ST, differed in the two paediatric groups evaluated. Previous corneal biomechanics studies in children did not show discordant characteristics compared with the general population in the same age group with an ORA, and so far, no studies have been conducted with Corvis ST [20,21].

In particular, in this study, HC deflection length, A1 deflection amplitude, and A1 deflection area were significantly lower in the MODY2 group (Table 4). According to these findings, there should be reduced corneal deformability in these eyes.

There were few correlations detected among the biomechanical features and the systemic parameters evaluated (Table 5), suggesting that corneal deformability could be positively associated to waist circumference (WC) and BMI. On the other hand, the lack of associations frequently observed in previous studies to support this theory suggests that diabetes does not significantly affect the corneal biomechanical properties in these patients. Thus, the different behavior of the cornea could be related to genetic alterations that affect this tissue, even in the absence of pathologic conditions.

In conclusion, this study shows, for the first time, some differences in corneal distortion features of MODY2 eyes, even if the data obtained need to be confirmed in a larger population and in studies including deeper correlation analyses between corneal biomechanical properties and systemic parameters of healthy subjects.

These findings can be of aid in suggesting new strategies to better understand the mechanisms underlying this rare pediatric disease. In particular, the correlations detected among corneal distortion parameters and WC, BMI, and HbA1c could be further investigated to identify novel biomarkers useful in both the early diagnosis and overall management of MODY2 patients.

## Figures and Tables

**Table 1 diagnostics-13-01500-t001:** Age, sex, and refractive characteristics of the overall study population.

		MODY2 ^1^ Group	Control Group	*p*-Value
AGE (years mean)		12.8 (±5.66)	13 (±5.05)	0.58
SEX (Males/Females)		8/7	7/8	
REFRACTIVE PARAMETERS:	UCVA ^2^	0.95 (±0.18)	0.89 (±0.22)	0.09
	BCVA ^3^	1	1	
	sphere (D)	−0.12 (±0.39)	−0.31 (±0.60)	0.62
	cylinder (D)	−0.05 (±0.20)	0	

^1^ MODY2 = Maturity Onset Diabetes of the Young type 2; ^2^ UCVA = Uncorrected visual acuity; ^3^ BCVA = Best-corrected visual acuity.

**Table 2 diagnostics-13-01500-t002:** Clinical characteristics of the overall study population.

		MODY2 ^1^ Group	Control Group
AUXOLOGICAL DATA (mean):	Weight (kg)	44.2 (±21.35)	44 (±29.40)
	Height (cm)	149.6 (±23.61)	126.5 (±59.75)
	BMI ^2^ (kg/mq)	18.62 (±4.65)	18.62 (±9.63)
Time elapsed since diagnosis (years)		8.33 (±6.52)	-
Waist circumference (cm)		64.57 (±11.95)	
Fasting blood sugar (mg/dL)		103.2 (±7.06)	-
HbA1c ^3^ value (%)		6.18 (±0.39)	-

^1^ MODY2 = Maturity Onset Diabetes of the Young type 2; ^2^ BMI = Body Mass Index; ^3^ HbA1c = glycosylated hemoglobin.

**Table 3 diagnostics-13-01500-t003:** Comparison of morphological parameters provided by Specular Microscope, Corvis ST, and Pentacam in MODY2 patients and in healthy subjects.

Instrument	Parameters	MODY2 ^1^ Group	Control Group	*p*-Value
Specular Microscope	ECD ^2^ (cd/mm^2^)	3083.70 ± 340.15	2934.23 ± 165.90	**0.039**
	CCT ^3^ (µm)	539.47 ± 27.64	558.15 ± 21.34	**0.006**
Corvis ST	IOP ^4^ (mmHg)	16.83 ± 2.98	16.83 ± 1.72	1.000
	Pachymetry (µm)	553.30 ± 26.70	574.58 ± 17.69	**0.001**
Pentacam	K1 F ^5^ (D)	43.40 ± 1.92	43.27 ± 0.98	0.753
	K2 F ^6^ (D)	44.30 ± 1.90	44.21 ± 1.09	0.826
	Km F ^7^ (D)	43.84 ± 1.89	43.73 ± 1.00	0.792
	Pachymetry Apex (µm)	558.61 ± 23.06	576.31 ± 17.72	**0.003**
	Pachymetry Pupil (µm)	557.93 ± 22.98	575.81 ± 18.05	**0.002**
	Corneal Volume 3 mm (mm^3^)	4.03 ± 0.17	4.17 ± 0.14	**0.002**
	Corneal Volume 5 mm (mm^3^)	11.76 ± 0.46	12.20 ± 0.36	**0.000**

^1^ MODY2 = Maturity Onset Diabetes of the Young type 2; ^2^ ECD = endothelial cell density; ^3^ CCT = central corneal thickness; ^4^ IOP = intraocular pressure; ^5^ K1 F = flattest keratometry; ^6^ K2 F = steepest keratometry; ^7^ Km F = mean keratometry.

**Table 4 diagnostics-13-01500-t004:** Comparison of biomechanical features provided by Corvis ST in MODY2 patients and in healthy subjects.

Parameters	MODY2 ^1^ Group	Control Group	*p* Value
Deformation Amplitude Max (mm)	1.01 ± 0.10	1.02 ± 0.09	0.851
A1 ^2^ Time (ms)	7.67 ± 0.36	7.66 ± 0.21	0.871
A1 ^2^ Velocity (m/s)	0.14 ± 0.02	0.14 ± 0.02	0.933
A2 ^3^ Time (ms)	21.54 ± 0.87	21.70 ± 0.53	0.391
A2 ^3^ Velocity (m/s)	−0.25 ± 0.04	−0.25 ± 0.03	0.399
HC ^4^ Time (ms)	17.09 ± 0.51	17.36 ± 0.53	0.059
Peak Distance (mm)	4.78 ± 0.32	4.84 ± 0.21	0.440
Radius (mm)	7.97 ± 1.16	8.43 ± 0.91	0.107
A1 ^2^ Deformation Amplitude (mm)	0.14 ± 0.01	0.14 ± 0.01	0.795
HC ^4^ Deformation Amplitude (mm)	1.01 ± 0.10	1.02 ± 0.09	0.851
A2 ^3^ Deformation Amplitude (mm)	0.47 ± 0.11	0.45 ± 0.06	0.497
A1 ^2^ Deflection Length (mm)	2.30 ± 0.12	2.37 ± 0.16	0.090
HC ^4^ Deflection Length (mm)	6.14 ± 0.49	6.44 ± 0.33	**0.010**
A2 ^3^ Deflection Length (mm)	3.47 ± 0.84	3.47 ± 0.67	0.996
A1 ^2^ Deflection Amplitude (mm)	0.10 ± 0.01	0.10 ± 0.01	**0.040**
HC ^4^ Deflection Amplitude (mm)	0.81 ± 0.10	0.82 ± 0.07	0.574
A2 ^3^ Deflection Amplitude (mm)	0.14 ± 0.12	0.13 ± 0.05	0.774
Deflection Amplitude Max (mm)	0.86 ± 0.18	0.84 ± 0.08	0.696
Deflection Amplitude Max (ms)	16.78 ± 2.98	16.01 ± 0.55	0.174
Whole Eye Movement Max (mm)	0.35 ± 0.07	0.33 ± 0.05	0.339
Whole Eye Movement Max (ms)	21.90 ± 1.29	21.62 ± 0.59	0.294
A1 ^2^ Deflection Area (mm^2^)	0.18 ± 0.02	0.20 ± 0.02	**0.002**
HC ^4^ Deflection Area (mm^2^)	2.82 ± 0.49	2.95 ± 0.35	0.265
A2 ^3^ Deflection Area (mm^2^)	0.36 ± 0.52	0.32 ± 0.18	0.696
Max Inverse Radius (mm^−1^)	0.17 ± 0.07	0.15 ± 0.02	0.151
Deflection Amplitude Ratio Max (2 mm)	4.00 ± 0.34	3.87 ± 0.32	0.150
Pachymety Slope (µm)	35.27 ± 5.40	39.46 ± 7.93	**0.033**
Deflection Amplitude Ratio Max (1 mm)	1.52 ± 0.04	1.50 ± 0.04	0.166
bIOP ^5^	16.74 ± 2.80	16.17 ± 1.56	0.353
Integrated Radius (mm^−1^)	7.09 ± 0.66	6.71 ± 0.73	0.055

^1^ MODY2 = Maturity Onset Diabetes of the Young type 2; ^2^ A1 = first applanation; ^3^ A2 = second applanation; ^4^ HC = highest concavity; ^5^ bIOP = biomechanically corrected intraocular pressure.

**Table 5 diagnostics-13-01500-t005:** Evaluation of correlations among the biomechanical characteristics of MODY2 corneas and the systemic parameters.

Best Model	Weight (kg)		Height (cm)		BMI ^1^ (kg/mq)		Waist Circumference (cm)		Fasting Blood Sugar (mg/dL)		HbA1c ^2^ (%)	
		*p*		*p*		*p*		*p*		*p*		*p*
Deformation Amplitude Max (mm)	0.019	0.984	0.013	0.991	0.005	0.982	62.315	<0.001	−0.001	0.997	7.35 × 10^−5^	0.997
A1 ^3^ Time (ms)	−0.007	0.987	0.002	0.997	−0.003	0.975	−0.009	0.969	−0.002	0.988	3.9 × 10^−5^	0.996
A1 ^3^ Velocity (ms)	0.056	0.989	0.029	0.995	0.015	0.986	449.089	<0.001	0.0133	0.992	−2.1 × 10^−5^	0.999
A2 ^4^ Time (ms)	−0.000	0.994	−0.0006	0.993	−4.4 × 10^−5^	0.997	−9.6 × 10^−5^	0.998	6.1 × 10^−5^	0.997	0.285	<0.001
A2 ^4^ Velocity (ms)	−0.010	0.995	−0.005	0.997	−0.004	0.991	−0.005	0.995	−0.004	0.993	4.49 × 10^−5^	0.999
HC ^5^ Time (ms)	−0.001	0.991	−0.001	0.995	−0.000	0.991	−0.000	0.995	−7.9 × 10^−5^	0.998	0.360	<0.001
Peak Distance (mm)	0.017	0.971	0.008	0.988	0.005	0.961	0.016	0.950	0.002	0.989	0.000	0.988
Radius (mm)	−0.000	0.993	−0.001	0.991	−4 × 10^−5^	0.997	−0.000	0.996	7.81 × 10^−5^	0.996	6.85 × 10^−6^	0.994
A1 ^3^ Deflection Amplitude (mm)	−0.026	0.995	−0.007	0.999	−0.008	0.993	−0.022	0.992	−0.005	0.997	−0.000	0.998
HC ^5^ Deflection Amplitude (mm)	0.0195	0.985	0.0135	0.991	0.005	0.982	62.316	<0.001	−0.001	0.997	7.35 × 10^−5^	0.997
A2 ^4^ Deflection Amplitude (mm)	0.001	0.999	0.0009	0.999	−4.6 × 10^−5^	0.999	0.004	0.989	−0.001	0.994	5.38 × 10^−5^	0.996
** *A1 ^3^ Deflection Length* ** * (**mm**)*	** *−0.004* **	** *0.992* **	** *−0.004* **	** *0.994* **	** *−0.001* **	** *0.992* **	** *−0.003* **	** *0.990* **	** *0.000* **	** *0.998* **	** *8.22 × 10^−6^* **	** *0.999* **
HC ^5^ Deflection Length (mm)	0.001	0.994	−0.000	0.999	0.000	0.989	10.121	<0.001	0.000	0.995	1.44 × 10^−5^	0.996
A2 ^4^ Deflection Length (mm)	9.57 × 10^−5^	0.999	−1.1 × 10^−5^	0.999	4.15 × 10^−5^	0.997	0.0002	0.995	−7.4 × 10^−5^	0.997	7.36 × 10^−7^	0.999
** *A1 ^3^ Deflection Amplitude* ** * (**mm**)*	** *−0.042* **	** *0.996* **	** *−0.026* **	** *0.998* **	** *−0.010* **	** *0.995* **	** *−0.054* **	** *0.991* **	** *0.011* **	** *0.997* **	** *−0.00097* **	** *0.995* **
HC ^5^ Deflection Amplitude (mm)	0.060	0.968	0.036	0.983	0.016	0.960	77.465	<0.001	0.002	0.996	−4.2 × 10^−5^	1.000
A2 ^4^ Deflection Amplitude (mm)	0.004	0.993	0.003	0.995	0.001	0.994	0.0027	0.991	−0.000	0.998	3.41 × 10^−6^	0.999
Deflection Amplitude Max (mm)	0.005	0.989	0.003	0.994	0.001	0.988	0.005	0.982	−0.001	0.996	−3E−06	1.000
Deflection Amplitude Max (ms)	−8.3 × 10^−5^	0.996	−9.5 × 10^−5^	0.996	−2 × 10^−5^	0.996	−2.1 × 10^−5^	0.998	−5 × 10^−5^	0.994	−7.2 × 10^−7^	0.998
Whole Eye Movement Max (mm)	−0.009	0.992	−0.008	0.994	−0.002	0.990	0.003	0.996	−0.003	0.991	0.000125	0.994
Whole Eye Movement Max (ms)	−0.000	0.997	−0.001	0.993	−2.3 × 10^−5^	0.999	0.000	0.991	−0.000	0.988	3.85 × 10^−6^	0.997
** *A1 ^3^ Deflection Area* ** * (**mm**)*	** *−0.0137* **	** *0.996* **	** *−0.011* **	** *0.997* **	** *−0.002* **	** *0.996* **	** *−0.008* **	** *0.996* **	** *−0.001* **	** *0.999* **	** *−0.00029* **	** *0.995* **
HC ^5^ Deflection Area (mm)	0.006	0.979	0.003	0.991	6.151	<0.001	22.222	<0.001	0.000	0.995	1.67 × 10^−5^	0.997
A2 ^4^ Deflection Area (mm)	0.001	0.994	0.001	0.995	0.000	0.995	0.001	0.991	−0.000	0.998	1.41 × 10^−6^	1.000

^1^ BMI = Body Mass Index; ^2^ HbA1c = glycosylated hemoglobin; ^3^ A1 = first applanation; ^4^ A2 = second applanation; ^5^ HC = highest concavity.

## Data Availability

Data presented in the manuscript are available from the corresponding authors upon reasonable request.

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
