# Peer review of "Analysis of Corneal Deformation in Paediatric Patients Affected by Maturity Onset Diabetes of the Young Type 2"

_diagnostics, 2023, doi:10.3390/diagnostics13081500_

Round 1

Reviewer 1 Report

The paper needs extensive editing

1. In abstract section : To evaluate for the first time corneal deformations in Maturity Onset Diabetes of the Young type 2 (MODY 2) paediatric subjects with a Scheimpflug based device, aiming to find both new biomarkers for MODY2 disease and both to better understand the pathogenesis of the disease

the sentence is too long to comprehend, and there are typo errors

2. It is not clear what the authors are trying to address in this paper

Author Response

Thank you for the time you spent in evaluating this manuscript and to provide us the opportunity to improve it. Changes requested are highlighted in yellow in the manuscript.

The paper needs extensive editing

Thank you for this suggestion. The whole manuscript has been edited by a native English scientific editor aiming to improve the overall quality of it.

  1. In abstract section: To evaluate for the first time corneal deformations in Maturity Onset Diabetes of the Young type 2 (MODY 2) paediatric subjects with a Scheimpflug based device, aiming to find both new biomarkers for MODY2 disease and both to better understand the pathogenesis of the disease

the sentence is too long to comprehend, and there are typo errors

Thank you for this comment, we modified it aiming to provide clearer concepts (lines 17-19).

  1. It is not clear what the authors are trying to address in this paper

Thank you for the opportunity to clarify this. The purpose of this paper is to look for new ocular biomarkers useful in the diagnosis and management of the MODY 2 patients and to have more information to better understand the mechanisms underlying this disease (lines 70-75).

Reviewer 2 Report

This paper presents the first evaluation of corneal deformations in Maturity Onset Diabetes of the Young type 2 (MODY2) pediatric patients. The authors used the Pentacam HR EM-3000 Specular Microscope and Corvis ST devices to perform ophthalmic evaluations in a small group of MODY2 pediatric subjects and age-matched healthy individuals. They found significantly lower values of Highest Concavity (HC) deflection length, Applanation 1 (A1) deflection amplitude and A1 deflection in MODY2 patients. They also found significant positive correlations between body mass index (BMI) and HC deflection area and between waist circumference (WC) and the several parameters. A positive correlation was also found between the glycosylated hemoglobin level (HbA1c) and Applanation 2 time and HC time.

Although the used sample was of small size, this study is worth reporting as it is the first study on corneal deformations in MODY2 pediatric patients. However, there are issues that should be addressed before the manuscript is suitable for publication.

Concerning the data in Table 1: Are the differences in sphere between the two groups statistically significant? Maybe it would be better to present p values regarding the comparisons between groups, for the parameters UCVA and sphere.

In the Materials and Methods section, it is not clear if both eyes were observed per subject, or, instead, imaging was done in just one eye. This must be clarified: how many eyes were included in the study? In case both eyes were used, they were both considered?

In Table 4, the p value for the parameter “A1 deflection area” should be at bold since its value is lower than the significance level.

In the paragraph beginning at line 236, it is stated that “among the parameters with a significant difference between healthy and MODY2 eyes, only HC deflection length provided a significant, positive, correlation with waist circumference, whereas, A1 deformation amplitude and A1 deformation area did not”. There must be some confusion. According to Table 4, A1 deformation amplitude did not differ significantly between patients and controls and there is no parameter named “A1 deformation area”. Should it be “A1 deflection area”?

The readability of Table 5 would benefit of evidencing (using bold, for example) the cases of statistically significant correlation.

In line 188 it is stated that no significant differences have been detected when evaluating the endothelial cell density. However, in Table 3 it is presented a p value of 0.039 for the ECD parameter and it is marked as a statistically significant difference. What is correct? In case the ECD differences are considered significant they should be discussed since several studies decreased endothelial cell density in Type 1 and Type 2 DM patients (see, for example, the review paper in 10.1016/j.visres.2017.03.002) and the present study reports a higher ECD for the MODY2.

The authors explain the result concerning central corneal thickness (CCT) with the small increase in HbA1c levels in pediatric MODY2 patients. They acknowledge that while this study found a lower CCT in the MODY2 group, different published studies regarding DM patients found higher CCT values that, in the pediatric population, are strongly influenced by HbA1c levels with a positive correlation. The discussion of this point could be enhanced since the small increase in the HbA1c levels of MODY2 patients can justify a lower-than-expected increase or no difference in the CCT of these patients, but not a decrease in this parameter. It would be useful to know the effect size or the minimum detectable difference for the statistical comparison of CCT between the MODY2 and the healthy groups.

The major concern I have regards the correlations found between biomechanical characteristics of MODY2 corneas and systemic parameters. If I understood correctly, correlations were only tested in the MODY2 group. So, is it possible that similar correlations could also be found in the healthy group? I am afraid that the final statement of the manuscript (“In particular, the correlations detected among corneal distortion parameters and WC, BMI and HbA1c could be deeply investigated in further studies in order both to establish new biomarkers useful in the early diagnosis and the overall management of the MODY2 patients”) is not fully supported by the experimental data since the study design does not allow to claim that the found correlations are a feature of MODY2 corneas and, therefore, have the potential to establish biomarkers for early diagnosis and management of these patients. The authors should comment on this. If such data is available, correlations should also be reported for the healthy group, particularly those found significant for the MODY2 patients.

Minor issues

The English usage is adequate although its quality decreases slightly in the Discussion section. This section also has a few typo errors. I suggest reviewing the English of that section. Some examples:

Line 267 “structure chances” should be structure changes.

Line 273 “distortion properties was still open”. I believe that discussion is still open.

Line 276-277 “Corneal biomechanical properties assessed”. It seems the word “were” is missing.

Also, in the abstract, the first sentence should be improved. One “both” is enough but may be it is better to remove the two.

Author Response

This paper presents the first evaluation of corneal deformations in Maturity Onset Diabetes of the Young type 2 (MODY2) pediatric patients. The authors used the Pentacam HR EM-3000 Specular Microscope and Corvis ST devices to perform ophthalmic evaluations in a small group of MODY2 pediatric subjects and age-matched healthy individuals. They found significantly lower values of Highest Concavity (HC) deflection length, Applanation 1 (A1) deflection amplitude and A1 deflection in MODY2 patients. They also found significant positive correlations between body mass index (BMI) and HC deflection area and between waist circumference (WC) and the several parameters. A positive correlation was also found between the glycosylated hemoglobin level (HbA1c) and Applanation 2 time and HC time.

Although the used sample was of small size, this study is worth reporting as it is the first study on corneal deformations in MODY2 pediatric patients. However, there are issues that should be addressed before the manuscript is suitable for publication.

Thank you for the time spent in evaluating this manuscript and to provide us the opportunity to improve it. Changes requested are highlighted in yellow in the manuscript.

Concerning the data in Table 1: Are the differences in sphere between the two groups statistically significant? Maybe it would be better to present p values regarding the comparisons between groups, for the parameters UCVA and sphere.

Thank you for this suggestion, we modified table 1 providing p values that showed no significant difference both in age and sphere and UCVA (table 1)

In the Materials and Methods section, it is not clear if both eyes were observed per subject, or, instead, imaging was done in just one eye. This must be clarified: how many eyes were included in the study? In case both eyes were used, they were both considered?

Thank you for this comment and for the opportunity to clarify this. Both eyes of both patients and controls were included in the study and in the statistical analysis. Aiming to accommodate the inner inter-eye correlation generalized estimating equation (GEE) was applied (lines 139-140 and 147-148).

In Table 4, the p value for the parameter “A1 deflection area” should be at bold since its value is lower than the significance level.

Thank you for this comment, we are sorry for this forgetfulness we modified table 4 as suggested.

In the paragraph beginning at line 236, it is stated that “among the parameters with a significant difference between healthy and MODY2 eyes, only HC deflection length provided a significant, positive, correlation with waist circumference, whereas, A1 deformation amplitude and A1 deformation area did not”. There must be some confusion. According to Table 4, A1 deformation amplitude did not differ significantly between patients and controls and there is no parameter named “A1 deformation area”. Should it be “A1 deflection area”?

Thank you for this comment, we are sorry for these confusing errors, we meant A1 deflection amplitude and A1 deflection area. We modified the text (line 229)

The readability of Table 5 would benefit of evidencing (using bold, for example) the cases of statistically significant correlation.

Thank you for this comment, we evidenced the significant parameters, making them bold, italics and bigger

In line 188 it is stated that no significant differences have been detected when evaluating the endothelial cell density. However, in Table 3 it is presented a p value of 0.039 for the ECD parameter and it is marked as a statistically significant difference. What is correct? In case the ECD differences are considered significant they should be discussed since several studies decreased endothelial cell density in Type 1 and Type 2 DM patients (see, for example, the review paper in 10.1016/j.visres.2017.03.002) and the present study reports a higher ECD for the MODY2.

Thanks for this comment, we added the details regarding the ECD in the text in the Results section, line 177 and discussed it in the Discussion section, lines 302-309, moreover proper references have been added.

The authors explain the result concerning central corneal thickness (CCT) with the small increase in HbA1c levels in pediatric MODY2 patients. They acknowledge that while this study found a lower CCT in the MODY2 group, different published studies regarding DM patients found higher CCT values that, in the pediatric population, are strongly influenced by HbA1c levels with a positive correlation. The discussion of this point could be enhanced since the small increase in the HbA1c levels of MODY2 patients can justify a lower-than-expected increase or no difference in the CCT of these patients, but not a decrease in this parameter. It would be useful to know the effect size or the minimum detectable difference for the statistical comparison of CCT between the MODY2 and the healthy groups.

Thank you for this comment, we modified the text related to the discussion of this aspect, aiming to clarify it and a reference has been added (lines 295-301).

The major concern I have regards the correlations found between biomechanical characteristics of MODY2 corneas and systemic parameters. If I understood correctly, correlations were only tested in the MODY2 group. So, is it possible that similar correlations could also be found in the healthy group? I am afraid that the final statement of the manuscript (“In particular, the correlations detected among corneal distortion parameters and WC, BMI and HbA1c could be deeply investigated in further studies in order both to establish new biomarkers useful in the early diagnosis and the overall management of the MODY2 patients”) is not fully supported by the experimental data since the study design does not allow to claim that the found correlations are a feature of MODY2 corneas and, therefore, have the potential to establish biomarkers for early diagnosis and management of these patients. The authors should comment on this. If such data is available, correlations should also be reported for the healthy group, particularly those found significant for the MODY2 patients.

 Thank you for this comment, the correlations between corneal biomechanical properties and systemic parameters were not evaluated because the latter were not available. We agree with you that this could have been a very interesting part of the study and we plan to design another one including them, this is why we included the sentence related to the “further studies”. We added a sentence aiming to clarify this (lines 342-343).

Minor issues

The English usage is adequate although its quality decreases slightly in the Discussion section. This section also has a few typo errors. I suggest reviewing the English of that section. Some examples:

Line 267 “structure chances” should be structure changes.

Line 273 “distortion properties was still open”. I believe that discussion is still open.

Line 276-277 “Corneal biomechanical properties assessed”. It seems the word “were” is missing.

Also, in the abstract, the first sentence should be improved. One “both” is enough but may be it is better to remove the two.

Thank you for these comments, we modified the text according to your suggestions.

Reviewer 3 Report

This manuscript does an excellent job analyzing for the first timethe corneal deformation in MODY 2 pediatric patients with a Scheimpflug based device. The authors found, for the first time, that there are differencies of corneal distortion features in the MODY 2 population, which brings originality to this research.

The title is appropriate for the content of the article. The abstract is concise and accurately summarizes the essential information of the paper. The manuscript is written in a clear, direct and active style, free from grammatical errors and other linguistic inconsistencies. 

The main strength of this paper is that it describes an actual problem and I appreciate the authors’ patience and professionalism in dealing with this situation. Moreover, this article analasys for the first time the corneal distortion features in the MODY 2 pediatric patients and this represents a very important novelty in the scientific field.

One possible criticism could be that the research was only made on 15 patients and data obtained needs to be confirmed in a larger population.

Author Response

This manuscript does an excellent job analyzing for the first timethe corneal deformation in MODY 2 pediatric patients with a Scheimpflug based device. The authors found, for the first time, that there are differencies of corneal distortion features in the MODY 2 population, which brings originality to this research.

The title is appropriate for the content of the article. The abstract is concise and accurately summarizes the essential information of the paper. The manuscript is written in a clear, direct and active style, free from grammatical errors and other linguistic inconsistencies.

The main strength of this paper is that it describes an actual problem and I appreciate the authors’ patience and professionalism in dealing with this situation. Moreover, this article analasys for the first time the corneal distortion features in the MODY 2 pediatric patients and this represents a very important novelty in the scientific field.

One possible criticism could be that the research was only made on 15 patients and data obtained needs to be confirmed in a larger population.

Thank you for the time you spent in evaluating this manuscript and for the kind evaluation. We agree with you that 15 patients is a small cohort of study but unfortunately this is a very rare disease in our area and we are not been able to enroll more patients.

Round 2

Reviewer 2 Report

The authors addressed properly my comments and the manuscript is now suitable for publication.

This is a preliminary study. The correlations between biomechanical characteristics and systemic parameters that were found on MODY2 corneas were not tested in healthy controls. Therefore, it is not possible to claim that these correlations are a feature of MODY2 and may serve as basis for biomarkers for early diagnosis and disease management. However, even with this limitation, the obtained results deserve to be reported.